# Regulation of Alcohol and Acetaldehyde Metabolism by a Mixture of *Lactobacillus* and *Bifidobacterium* Species in Human

**DOI:** 10.3390/nu13061875

**Published:** 2021-05-30

**Authors:** Su-Jin Jung, Ji-Hyun Hwang, Eun-Ock Park, Seung-Ok Lee, Yun-Jo Chung, Myung-Jun Chung, Sanghyun Lim, Tae-Joong Lim, Yunhi Ha, Byung-Hyun Park, Soo-Wan Chae

**Affiliations:** 1Clinical Trial Center for Functional Foods, Jeonbuk National University Hospital, Jeonju 54907, Jeonbuk, Korea; sjjeong@jbctc.org (S.-J.J.); ctv5202@gmail.com (J.-H.H.); eopark@jbctc.org (E.-O.P.); 2Biomedical Research Institute, Jeonbuk National University Hospital, Jeonju 54907, Jeonbuk, Korea; solee@jbnu.ac.kr (S.-O.L.); yjchong@jbnu.ac.kr (Y.-J.C.); 3Division of Gastroenterology and Hepatology, Department of Internal Medicine, Jeonbuk National University Medical School, Jeonju 54896, Jeonbuk, Korea; 4R&D Center, Cell Biotech, Co., Ltd., Gimpo 10003, Gyeongi, Korea; ceo@cellbiotech.com (M.-J.C.); shlim@cellbiotech.com (S.L.); tjlim@cellbiotech.com (T.-J.L.); 5Clinical Research and Development Team, Cell Biotech, Co., Ltd., Gimpo 10003, Gyeongi, Korea; yhha@cellbiotech.com; 6Department of Biochemistry and Molecular Biology, Jeonbuk National University Medical School, Jeonju 54896, Jeonbuk, Korea

**Keywords:** probiotics, *Lactobacillus*, *Bifidobacterium*, alcohol, acetaldehyde, *ALDH2* gene

## Abstract

Excessive alcohol consumption is one of the most significant causes of morbidity and mortality worldwide. Alcohol is oxidized to toxic and carcinogenic acetaldehyde by alcohol dehydrogenase (ADH) and further oxidized to a non-toxic acetate by aldehyde dehydrogenase (ALDH). There are two major ALDH isoforms, cytosolic and mitochondrial, encoded by *ALDH1* and *ALDH2* genes, respectively. The *ALDH2* polymorphism is associated with flushing response to alcohol use. Emerging evidence shows that *Lactobacillus* and *Bifidobacterium* species encode alcohol dehydrogenase (ADH) and acetaldehyde dehydrogenase (ALDH) mediate alcohol and acetaldehyde metabolism, respectively. A randomized, double-blind, placebo-controlled crossover clinical trial was designed to study the effects of *Lactobacillus* and *Bifidobacterium* probiotic mixture in humans and assessed their effects on alcohol and acetaldehyde metabolism. Here, twenty-seven wild types (*ALDH2*1/*1*) and the same number of heterozygotes (*ALDH2*2/*1*) were recruited for the study. The enrolled participants were randomly divided into either the probiotic (Duolac ProAP4) or the placebo group. Each group received a probiotic or placebo capsule for 15 days with subsequent crossover. Primary outcomes were measurement of alcohol and acetaldehyde in the blood after the alcohol intake. Blood levels of alcohol and acetaldehyde were significantly downregulated by probiotic supplementation in subjects with *ALDH2*2/*1* genotype, but not in those with *ALDH2*1/*1* genotype. However, there were no marked improvements in hangover score parameters between test and placebo groups. No clinically significant changes were observed in safety parameters. These results suggest that Duolac ProAP4 has a potential to downregulate the alcohol and acetaldehyde concentrations, and their effects depend on the presence or absence of polymorphism on the *ALDH2* gene.

## 1. Introduction

Chronic alcohol consumption is one of the major causes of morbidity and mortality, ranging from simple steatosis to hepatocellular carcinoma [1]. Once ingested, alcohol is oxidized to toxic and carcinogenic acetaldehyde by alcohol dehydrogenase (ADH) and further oxidized to a non-toxic acetate by aldehyde dehydrogenase (ALDH) [2]. There are two major ALDH isoforms, cytosolic ALDH1 and mitochondrial ALDH2. Most Caucasians have two isozymes, while approximately 30–50% of East Asians have ALDH2 deficiency that results from the inheritance of the mutant *ALDH2*2* allele [3]. As the *ALDH2*2* allele has very low enzymatic activity, alcohol ingestion in subjects with one or both alleles of *ALDH2*2* causes a marked elevation of blood acetaldehyde levels, which are known to generate dysphoric effects that lead to an aversion to ethanol [4].

Probiotics are microorganisms that can change the gut lumen favoring an anti-inflammatory milieu, resulting in decreased pathogenic bacterial toxins and improved barrier integrity. *Lactobacilli* and *Bifidobacteria* are important members of the indigenous flora of the large intestine in humans and are also the best characterized and the most commercialized probiotics. The therapeutic potential of these probiotics on alcohol-induced liver diseases has been reported in animal and human studies [5,6,7,8]. Recently, Lu et al. [9] showed that *Bacillus subtilis* co-expressing ADH and ALDH has a protective effect against the development of alcohol-induced liver damage in mice, suggesting that probiotics also play a key role in alcohol intoxication. However, no study has been conducted to evaluate whether probiotics influence alcohol metabolism in humans. Thus, in this investigation, a randomized, double-blind, placebo-controlled crossover study was performed to assess the capacity of *Lactobacilli* and *Bifidobacteria* to improve alcohol metabolism. Moreover, their role in reducing hangover symptoms with respect to genetic variations of *ALDH2* was investigated.

## 2. Materials and Methods

### 2.1. Test Supplements

Duolac ProAP4 constitutes four probiotics [*Lactobacillus gasseri* CBT LGA1, *Lactobacillus casei* CBT LC5, *Bifidobacterium lactis* CBT BL3, and *Bifidobacterium breve* CBT BR3] and manufactured by Cell Biotech (Gimpo, Gyeonggi, Korea) [10]. It is double-coated and contained over 125,000,000 CFU/400 mg capsule of probiotics. Placebo was made of fructo-oligosaccharide and dextrose and had the same appearance, flavor, and weight as the Duolac ProAP4. According to the Ministry of Food and Drug Safety (MFDS) of Korea, intake of probiotics in healthy functional foods is 1 × 10^8^~10^10^ CFU per daily serving. Previous pre-clinical studies show that serum alcohol and serum acetaldehyde concentrations were notably decreased in animals receiving Duolac ProAP4 administration [10]. Based on these results, the appropriate probiotic dose for subjects in the present study was 5 × 10^8^ CFU/day.

### 2.2. Subjects

This study was performed from 11 March to 26 October 2019 after receiving approval from the Institutional Review Board (IRB) of Jeonbuk National University Hospital (IRB No. JUH 2018-12-019). The entire study was conducted in accordance with the provisions of the Helsinki Declaration and the provisions of the Korean Good Clinical Practice (KGCP). The study was registered in the Clinical Research Information Service of Korea (Approval number: KCT0005361). All participants were instructed to take four whitening hard capsules per day (two capsules each after breakfast and dinner). Duolac ProAP4 and placebo capsules were packaged indistinguishably and labeled with a serial number. Participants were instructed to bring all the remaining supplements at each visit and were withdrawn from the study if the supplement consumption was <80% of the prescribed dose. Alcohol challenge test was carried out on the first period (day 15) and second period (day 58); after 30 min of standard meal intake. All participants consumed the day’s supplements (four capsules/day) with water.

The participants were recruited by advertising the investigation through various methods like brochures, posters, and JUH website. A total of 94 participants were eligible after screening tests such as questionnaires, physical examinations, genetic tests, and laboratory examinations. Participants were enrolled within four weeks after the screening test. Prior to the trial, informed consent was obtained from all the participants.

Inclusion criteria were as follows: (1) male aged ≥19 and ≤65 years at the time of the screening test, (2) body mass index (BMI) of 18 to 25 kg/m^2^, (3) healthy adults with post-drinking hangover experience and those who had fully understood the detailed description of the study and voluntarily agreed to participate. Exclusion criteria for the study were: (1) a person who is a homozygote type (*ALDH2*2/*2*) of the *ALDH2* genotype, (2) a person who is hypersensitive or has a history of clinically significant hypersensitivity to drugs, alcohol, products, or other ingredients, (3) a person who has taken a drug that induces and inhibits drug metabolic enzymes, such as barbital drugs, within one month from the date of screening test, (4) a person who has taken drugs that affect the clinical results such as alcohol metabolism within one month from the screening test (drugs with a risk of gastrointestinal bleeding such as aspirin, antipyretic analgesics, anti-inflammatory analgesics, antibiotics, herbal medicines, oral steroids, hormones, etc.), (5) a person who has taken drugs, products, and health functional foods that are believed to affect the intestines, such as probiotics, *Lactobacillus* drinks (e.g., yogurt), and dairy products, within one month from the date of screening test, (6) a person who has taken drugs, products, and health functional foods that are believed to have an effect on the stomach and liver, such as milk thistle (silymarin) and licorice extract, within one month from the date of screening test, (7) a person who has taken drugs, products, and health functional foods that are deemed unsuitable for participation in the study by the person in charge of the study, such as hangover relief products, (8) A person who has consumed excessive alcohol within one week from the screening test date, (9) a person with severe acute or chronic cardiovascular diseases, metabolic diseases, liver and biliary diseases, pancreatic diseases, muscle diseases, neurological diseases, mental disorders, endocrine diseases, immune diseases, kidney diseases, malignant tumors, lung diseases, and other diseases requiring treatment, (10) a person who has or is undergoing treatment for a clinically significant gastrointestinal disease such as gastric or duodenal ulcer, (11) a person who has a history of a gastrointestinal disease such as Crohn’s disease or gastrointestinal surgery (excluding simple appendectomy or herniotomy) that could affect the absorption of the study diet, (12) a person who has received antipsychotic drug within 2 months from the date of the screening test, (13) a person who has or is suspected of having a history of alcoholism or drug abuse, (14) a person who has participated in other studies within 3 months from the screening test date (except simple observational studies in which there was no intra-body administration of drugs or foods (injection, ingestion, insertion, etc.)), (15) a person who has donated whole blood within 2 months from the date of screening or donated apheresis within 2 weeks from the date of screening, (16) a person who has serum AST, ALT, or creatine kinase levels two times greater than the upper limit of the reference range or serum creatinine level over 2.0 mg/dL in diagnostic tests, and (17) a person who is deemed unfit for this study by the tester due to diagnostic test results or other reasons.

### 2.3. Genotyping

The *ALDH2* gene was classified as wild type (*ALDH2*1/*1*), homozygote type (*ALDH2*2/*2*), and heterozygote type (*ALDH2*2/*1*) through single nucleotide polymorphism (SNP) r671 analysis. The variant *ALDH2*2* type was caused by a single-point mutation (G-A) of exon 12, which induces amino acid substitution from glutamine to lysine (E487K).

### 2.4. Study Design

The study was designed as a randomized, double-blind, and placebo-controlled crossover trial. Participants who met the entry criteria and responded via a telephone screening interview were scheduled for a baseline visit. The evaluation included a physical test, electrocardiogram, and blood parameters. After obtaining the written informed consent, 54 participants were assigned to either group A (Duolac ProAP4 intake → washout → placebo intake) or group B (placebo intake → washout → Duolac ProAP4 intake). Alcohol challenge test was performed after an overnight fast on day 15 and day 58. This study was conducted as a crossover trial to reduce the differences between individuals. During the trial period, between the first period (day 15) and the second period (day 58), a 28-day wash-out was placed to exclude the carryover effect of the test products (Figure 1). The participants were asked to maintain their diet during the study period and avoid eating any related health functional foods or dietary supplements. During the study period, all subjects were educated to refrain from drinking alcohol. Subjects were educated to record the frequency of drinking alcohol consumed by themselves during the test period once a week on the alcohol consumption frequency surveys distributed in the first visit and third visit. At the second and fourth visit, an alcohol breath test was performed using a breathalyzer (ALCOFIND DA-5000, DA TECH Co., Ltd., Incheon, Korea) to confirm alcohol consumption before participating in the alcohol challenge test. Participants were also asked to report any adverse events or any changes in training, lifestyle, eating patterns, and pill compliance.

### 2.5. Alcohol Challenge Test

All subjects fasted for at least 12 h before the start of the trial and were asked to consume the same standard diet (low-fat diet) on the morning of the day of the clinical study. The standard diet is made up of brown rice, Chinese cabbage soup, side dishes (Korean beef stew, egg roll, and seasoned mung bean jelly salad, seasoned spinach), and sliced radish kimchi. The standard diet has about 700 Kcal. Nutritional information for the standard diet was analyzed using CAN-Pro 4.0^®^ software (The Korean Nutrition Society, Seoul, Korea). Subjects took test products (either Duolac Pro AP4 or placebo) 30 min after eating a standard meal. During the alcohol challenge test, participants had a meal (standard diet) with alcohol (40% *v*/*v*, Absolut Vodka, The Absolut Company AB, Stockholm, Sweden). Alcohol consumed with water at 1:1 ratio amounting to 0.8 g per kg body weight of the study participants and consumed within 30 min with a small amount of snack. Body weight was based on the measurements of the first and third visit. Blood levels of alcohol and acetaldehyde were measured at 0, 0.5, 1, 2, 4, and 6 h after alcohol drinking.

### 2.6. Outcome Measurements

#### 2.6.1. Primary Outcomes

The primary outcomes were alcohol and acetaldehyde concentrations in the blood after the alcohol intake. Blood samples were obtained in anticoagulating tubes containing potassium-EDTA (BD Biosciences, San Jose, CA, USA) at baseline and at 0, 0.5, 1, 2, 4, and 6 h after the alcohol administration. Blood alcohol concentration was detected by headspace gas chromatography with flame ionization detection (HS-GC-FID) [11]. A 100 μL of whole blood was diluted with 1000 μL of internal standard solution in each vial. The samples were determined on a HS-GC-FID system (6890GC-FID, Agilent Technologies, Santa Clara, CA, USA) with headspace autosampler (G1888A, Agilent Technologies, Santa Clara, CA, USA). The conditions of analysis were as follows: DB-624 column (30 m × 0.251 mm × 1.40 mm; Agilent Technologies, Wilmington, DE, USA); 0–30 min, oven temperature program (40 °C for 3 min hold, 10 mL/min up to 260 °C, 5 min hold); headspace oven temperature, 80 °C; sample heating time, 15 min.

Blood acetaldehyde concentration was detected by liquid chromatography-tandem mass spectrometry (LC-MS/MS) with multiple reaction monitoring (MRM) [12]. Briefly, 1000 μL of whole blood was added in each vial containing mixture of 1 mL of saturated sodium nitrite and 100 μL of acetone for the internal standard. After adding 2, 4-dinitrophenylhydrazine (DNPH) cartridge, the mixtures were reacted for 24 h in the dark condition. The samples were extracted with 1 mL acetonitrile and detected using 6410 Triple Quad LC-MS/MS (Agilent Technologies, Wilmington, NC, USA). The analytical HPLC column was a reverse phase column (Shiseido CAPCELL, C18, 5 um, 2.0 mm × 10 cm). The flow rate was 0.23 mL/min and the elution was done with a gradient of water and acetonitrile containing 0.1% formic acid. Fragmentor voltage and collision voltage were set at 100 V and 10 V. Detection of the ions was carried out with MRM by monitoring the transition pairs of m/z 225.1 → 208.3 (aldehyde-DNPH). Data acquisition was performed with the MassHunter Software (Version B.04.00, Agilent, Santa Clara, CA, USA). At the same time point, expiratory alcohol concentration was measured by Lion SD-400 Breath Alcohol Analyser (Lion Laboratories, Barry, Vale of Glamorgan, UK). Maximum plasma concentration (C_max_), the time to reach it C_max_ (T_max_), and the incremental area under the curve (iAUC) were calculated using the concentrations of alcohol and acetaldehyde in the blood, and trapezoidal method was used for calculating the iAUC used [13].

#### 2.6.2. Secondary Outcomes

The secondary outcomes were Alcohol Hangover Questionnaire (AHQ), liver function test, and blood glucose levels. AHQ was conducted within 8 h of alcohol consumption during the alcohol challenge test. AHQ consisted of 20 questions, including questions about thirst, sleepiness, headache, dizziness, vomiting, helplessness, abdominal pain, diarrhea, concentration difficulty, and sensitivity to irritation [14]. Each symptom is rated on a 5-point Likert scale ranging from 1 (no symptom) to 5 (extremely severe symptom). The total points range from 20 to 100. Liver enzymes tests (AST, ALT, ALP, and γ-GT) were measured at 0, 1, and 6 h after the alcohol consumption. Blood glucose levels were measured at 0 and 6 h after drinking alcohol.

### 2.7. Safety Outcome Measurements

At each visit, participants underwent electrocardiogram, laboratory tests (WBC, RBC, Hb, Hct, platelet, ALP, γ-GT, AST, ALT, total bilirubin, total protein, albumin, BUN, creatinine, creatine kinase, total cholesterol, triglyceride, glucose, and hs-CRP), and vital signs (systolic blood pressure, diastolic blood pressure, and pulse) for safety evaluation. WBC, RBC, Hb, Hct, and platelet were measured using automated hematology analyzer XE-5000TM (Sysmex, Kobe, Japan). ALP, γ-GT, AST, ALT, total bilirubin, total protein, albumin, BUN, creatinine, creatine kinase, total cholesterol, triglyceride, glucose, and hs-CRP were measured using the ADVIA^®^ 2400 chemistry system (Siemens, Munich, Germany).

### 2.8. Evaluation of Diet and Physical Activity

Three-day food and physical activity records were collected at each visit to evaluate the usual diet and physical activity patterns of the participants. Dietary intake was analyzed by the same dietitian using CAN-pro 4.0 software (The Korean Nutrition Society, Seoul, Korea), and physical activity was assessed using a metabolic equivalent task (MET) assessment using the global physical activity questionnaire (GPAQ) developed by the World Health Organization [15].

### 2.9. Sample Size

Sample size was calculated to detect the blood acetaldehyde AUC changes 0.008 ± 0.023 mg∙h/dL between the Duolac ProAP4 and placebo groups. The sample size required to maintain 80% statistical power at a 5% significance level (two-tailed test) was calculated to be 40 persons per group. Therefore, a total of 54 people was required, assuming a dropout ratio of 25%.

### 2.10. Statistical Analysis

Statistical analysis was performed using the SAS version 9.4 (SAS Institute, Charlotte, NC, USA). Analyses were performed according to intention-to-treat principles. For each variable, participants were grouped according to the sequence of intervention (Duolac ProAP4, then placebo or placebo, then Duolac ProAP4). The student’s paired *t*-test was used for continuous measurements to assess differences between before and after the 15-day intervention period. Fixed effects included treatment group, treatment visit, and the interaction between treatment group and visit. Data are shown as the mean ± standard deviation (SD). A *p*-value less than 0.05 was considered statistically significant.

## 3. Results

### 3.1. Demographic Characteristics of Participants

Among the 94 participants screened, 40 participants were excluded due to laboratory test results consistent with the exclusion criteria. The remaining 54 participants fulfilled the study criteria and included in the investigation. The supplement was consumed according to the order of intake of the assignment group, which was randomly assigned to either group A or group B (Group A: Duolac ProAP4, then placebo and Group B: placebo, then Duolac ProAP4). Moreover, the assigned group was stratified by the *ALDH2* genotypes. According to the crossover design, participants received the opposite treatment after a 28-day washout period. During the study participation period, six people in group A and eight people in group B violated the human application test plan, 40 participants (21 in group A and 19 in group B) were able to finish the study (Figure 2). Table 1 shows the general characteristics of the 54 participants. Baseline characteristics of age, height, weight, BMI, drinking, smoking, blood pressure, pulse, temperature, and thyroid-stimulating hormone (TSH) were not significantly different between the wild and heterozygote types.

### 3.2. Diet Intake and Physical Activity

Significant differences in dietary intakes (calories, carbohydrates, protein, fat, and fiber) or physical activity (MET) were not confirmed between the groups during the intervention period (data not shown).

### 3.3. Efficacy Evaluation

#### 3.3.1. Primary Outcome

Table 2 shows the variation in blood acetaldehyde concentration after 15 days of Duolac ProAP4 supplementation. In the heterozygote group, Duolac ProAP4 supplementation clearly accelerated alcohol metabolism as acetaldehyde concentrations at 0.5, 1, and 6 h after alcohol consumption, and C_max_, and iAUC were significantly lower in Duolac ProAP4 supplemented participants compared with those of placebo group (*p* < 0.05) (Appendix A). However, these effects were not observed in wild-type participants. Alcohol concentrations were higher in heterozygote group regardless of Duolac ProAP4 supplementation compared to those in wild-type group. To reiterate, Duolac ProAP4 supplementation significantly decreased the alcohol concentration in the heterozygote group compared to the placebo group (Table 3).

#### 3.3.2. Secondary Outcomes

The alcohol challenge test after ingestion of the test products in this study revealed a notable difference between the two groups as the ALP levels after and before alcohol consumption in the Duolac ProAP4 group decreased compared to the placebo group (*p* = 0.001). The analysis of the hetero-type group, the liver enzymes of AST (1 h), ALT (1 h), and ALP (6 h) in the Duolac ProAP4 group were significantly decreased compared to the placebo group (*p* < 0.05) (Table 4). The blood glucose levels before alcohol consumption (0 h) were significantly higher in the placebo group than in the Duolac ProAP4 group. In addition, modifications in the blood glucose levels after alcohol intake (6 h) tended to decrease more in the placebo group than the Duolac ProAP4 group. However, there was no significant difference between the groups (Table 4).

AHQ of hangover symptom index was measured within 8 h of the alcohol consumption (Appendix A). The sum of all the items in each AHQ, the sum of 13 major symptoms of hangover [16], and the sum of score of 7 items [14] were compared. In contrast to the changes of alcohol and acetaldehyde concentrations, there were no significant difference between the two groups in the total score, score of 13 major hangover symptoms, and score of 7 items.

### 3.4. Safety and Adverse Events

No serious adverse events were reported during the study period. The laboratory tests, electrocardiogram, and vital signs were in the normal range. Thus, no participants withdrew because of adverse events.

## 4. Discussion

Previously, Cell Biotech Co Ltd. has screened 19 CBT probiotic species of *Lactobacillus* and *Bifidobacterium* to choose the best combination of probiotic strains for alcohol detoxification. In that investigation, they found that *Lactobacillus gasseri* CBT LGA1, *Lactobacillus casei* CBT LC5*, Bifidobacterium lactis,* CBT BL3 and *Bifidobacterium breve* CBT BR3 were highly effective in alcohol metabolism [10]. Specifically, *Lactobacillus gasseri* CBT LGA1 and *Bifidobacterium lactis* CBT BL3 demonstrated a high capacity for ethanol metabolism, while *Lactobacillus casei* CBT LC5 and *Bifidobacterium breve* CBT BR3 accelerated acetaldehyde metabolism. Further, the mixture of these four probiotics (Duolac ProAP4) was observed to benefit acute alcohol toxicity in rats [10]. Here, we evaluated the effect of Duolac ProAP4 on alcohol detoxification in humans. Consistent with the animal study, this randomized placebo-controlled crossover study demonstrates that Duolac ProAP4 supplementation results in lower blood concentrations of alcohol and acetaldehyde in the heterozygote (*ALDH2*2/*1*) subjects, but not in wild-type *(ALDH2*1/*1*) subjects. These observations distinctly suggest that Duolac ProAP4 supplementation is an effective way to maintain lower alcohol and acetaldehyde concentrations in humans.

Previously, Cell Biotech Co Ltd. and other groups have shown that *Lactobacillus* and *Bifidobacterium* species encode ADH and ALDH [17,18,19]. In this study, Duolac ProAP4 supplementation significantly decreased plasma concentrations of acetaldehyde 1 h after the alcohol ingestion compared with those of placebo group (0.108 ± 0.063 mg/dL in Duolac ProAP4 group vs. 0.147 ± 0.092 mg/dL in placebo group, *p* = 0.005) cemented the previously observed notion. However, Duolac ProAP4 supplementation did not affect alcohol levels 30 min after the alcohol ingestion (81.34 ± 31.55 mg/dL in Duolac ProAP4 group vs. 77.28 ± 29.39 mg/dL in placebo group, *p* = 0.425). These results indicate that Duolac ProAP4 does not affect the alcohol breakdown and its absorption in the stomach instantly, but it accelerates acetaldehyde oxidation into acetate in the intestine. Interestingly, Duolac ProAP4 supplementation significantly decreased the blood concentrations of acetaldehyde 6 h after the alcohol ingestion compared with those of placebo group (0.003 ± 0.005 mg/dL in Duolac ProAP4 group vs. 0.005 ± 0.009 mg/dL in placebo group, *p* = 0.019). These results suggest that Duolac ProAP4 may also increase acetaldehyde metabolism in the liver. Probiotic products containing *Lactobacillus* and *Bifidobacterium* actively promoted alcohol metabolism where it rapidly decompose alcohol and metabolizes it to acetaldehyde, a harmful compound to the human body [17]. Previous studies have reported on the possibility of detoxification. In line with these studies, Cell Biotech Co Ltd. and others have shown that probiotics supplementation has positive effects, alleviating acute alcoholic liver injury [5,6,7,8,20].

Heterozygote subjects taking Duolac ProAP4 showed an evident suppression in alcohol and acetaldehyde concentrations over time. However, those changes were not found in the wild-type subjects. These observations are unexpected, and it is difficult to explain these findings from the viewpoint of Duolac ProAP4’s ALDH enzyme activity. One possible speculation is the difference in the gut microbiota community between the two groups. It is well documented that subjects with a single nucleotide polymorphism on *ALDH2* gene tended to avoid excess alcohol drinking because of unpleasant hangover symptoms secondary to the failure of acetaldehyde metabolism [3,4]. Differences in alcohol ingestion potentially affect the composition of bowel flora. Evidently, alcoholics demonstrated to have reduced numbers of *Lactobacilli*, *Bifidobacteria*, and *Enterococci*, while there is an increase in the population of *E. coli* [8]. Similarly, animal studies have also reported a strong association between alcohol consumption and bowel flora composition [21,22]. Indeed, when we carefully compared the alcohol drinking history, we found that although there was no statistical significance, heterozygote subjects took less amount of alcohol compared to the wild-type subjects. Meanwhile, our study showed that Duolac ProAP4 supplementation substantially reduced blood acetaldehyde levels but did not relieve hangover symptom scores in heterozygote and wild-type subjects. These findings are contrary to our expectation that acetaldehyde is the main contributor to the development of hangover symptoms. Under physiological conditions, venous acetaldehyde produced in the liver can’t reach the brain due to the high ALDH2 enzyme activity of the endothelial cells that line the blood brain barrier [23]. Therefore, Duolac ProAP4 may not affect acetaldehyde metabolism in the brain. However, as acetaldehyde is considered as a key player in many actions of ethanol in the brain, including behavioral changes, Duolac ProAP4 still represents a valuable therapeutic option in the management of the alcohol abuse disorders. Additionally, other factors like inflammatory cytokines, fluid imbalance, gender, ethnicity, genetic background, and nutritional status are associated with the frequency and severity of hangover symptoms along with blood acetaldehyde concentrations [24]. Thus, future studies are certainly needed to analyze the aforementioned parameters.

## 5. Conclusions

The present findings suggest that the mixture of four probiotics (Duolac ProAP4) is practically handy in the management of alcohol metabolism in the *ALDH2*2/*1* subjects. Moreover, study warrants a large-scale clinical study to test if Duolac ProAP4 could be used to treat individuals with hangover symptoms after alcohol drinking.

## Figures and Tables

**Figure 1 nutrients-13-01875-f001:**
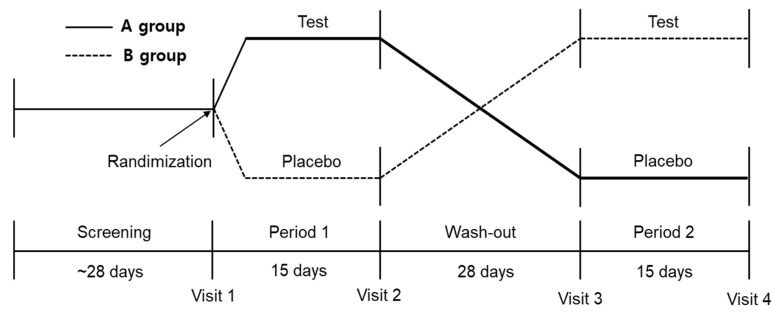
Scheme of the crossover design protocol.

**Figure 2 nutrients-13-01875-f002:**
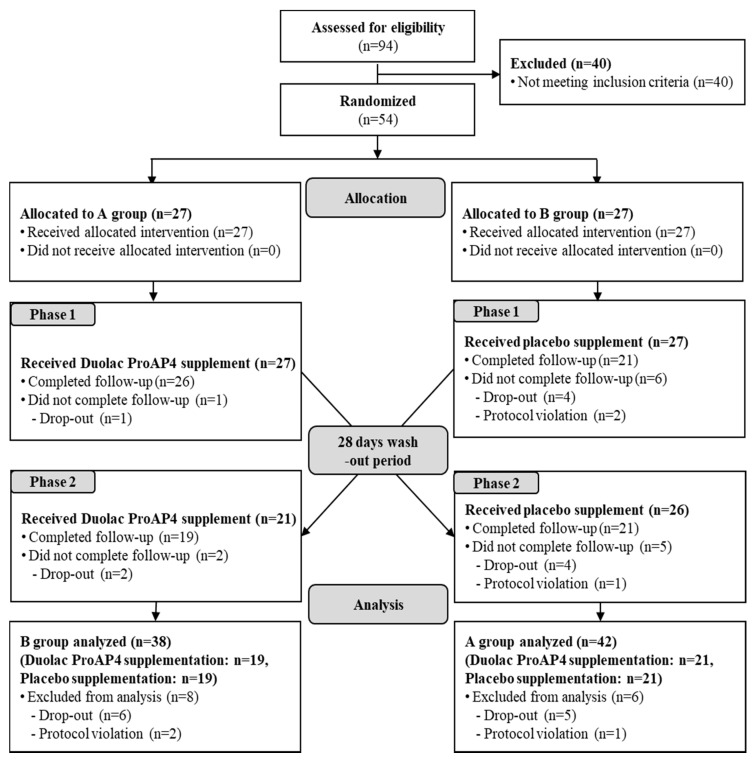
Flow diagram showing the selection and allocation of participants in the investigation.

**Table 1 nutrients-13-01875-t001:** Demographic characteristics of the study subjects.

Variables		Wild Type (*ALDH2*1/*1*, *n* = 27)	Heterozygote (*ALDH2*2/*1*, *n* = 27)	Total Group (*n* = 54)	
Age (years)		25.26 ± 2.61	24.89 ± 2.97	25.07 ± 2.77	
Height (cm)		176.15 ± 4.82	175.07 ± 5.27	175.61 ± 5.03	
Weight (kg)		70.61 ± 8.06	70.37 ± 8.03	70.49 ± 7.97	
Body mass index (kg/m^2^)		22.77 ± 2.14	22.93 ± 1.84	22.85 ± 1.98	
Drinking (yes/no)	non-drinker (*n*, %)	0, 0	0, 0	0, 0	
	past drinker (*n*, %)	0, 0	0, 0	0, 0	
	drinker (*n*, %)	27, 100	27, 100	54, 100	
Alcohol period (years)		6.00 ± 1.96	5.81 ± 2.32	5.91 ± 2.13	
Alcohol consumption (units/week)		7.38 ± 2.41	4.13 ± 2.22	5.75 ± 2.82	
Drinking within a week	yes (*n*, %)	23, 85	25, 93	48, 89	
	no (*n*, %)	4, 15	2, 7	6, 11	
Smoking	non-smoker (*n*, %)	17, 63	18, 67	35, 65	
	past smoker (*n*, %)	0, 0	0, 0	0, 0	
	Smoker (*n*, %)	10, 37	9, 33	19, 35	
Smoking period (years)		6.30 ± 2.71	3.67 ± 2.40	5.05 ± 2.84	
Smoking consumption (units/week)		10.10 ± 5.34	7.44 ± 4.69	8.84 ± 5.09	
Smoking within a week	yes (*n*, %)	10, 100	8, 89	18, 95	
	no (*n*, %)	0, 0	1, 11	1, 5	
Systolic blood pressure (mmHg)		119.81 ± 8.26	119.04 ± 10.36	119.43 ± 9.29	
Diastolic blood pressure (mmHg)		71.70 ± 8.88	70.48 ± 8.17	71.09 ± 8.47	
Pulse (BPM)		80.48 ± 9.93	72.56 ± 7.71	76.52 ± 9.67	
Temperature		36.2 ± 0.21	36.24 ± 0.24	36.22 ± 0.23	
Thyroid stimulating hormone		1.84 ± 1.32	1.73 ± 0.65	1.79 ± 1.03	

Values are presented as mean ± SD or frequency (%). Abbreviation: BPM, beats per minute.

**Table 2 nutrients-13-01875-t002:** Variation in blood acetaldehyde concentration flowing alcohol challenge test after 15 days of supplementation.

		Wild Type (*ALDH2*1/*1*)			Heterozygote (*ALDH2*2/*1*)			Total Group		
		Duolac ProAP4 Group (*n* = 19)	Placebo Group (*n* = 19)	*p*-Value ^(1)^	Duolac ProAP4 Group (*n* = 21)	Placebo Group (*n* = 21)	*p*-Value ^(1)^	Duolac ProAP4 Group (*n* = 40)	Placebo Group (*n* = 40)	*p*-Value ^(1)^
Blood acetaldehyde level (mg/dL)	0 h	0.000 ± 0.000	0.001 ± 0.002	0.117	0.000 ± 0.000	0.000 ± 0.001	0.553	0.000 ± 0.0002	0.001 ± 0.000	0.094
	0.5 h	0.007 ± 0.000	0.005 ± 0.006	0.660	0.113 ± 0.059	0.150 ± 0.085	0.018	0.063 ± 0.070	0.081 ± 0.096	0.040
	1 h	0.004 ± 0.010	0.005 ± 0.010	0.773	0.108 ± 0.063	0.147 ± 0.092	0.005	0.059 ± 0.070	0.080 ± 0.097	0.006
	2 h	0.002 ± 0.004	0.002 ± 0.004	0.941	0.050 ± 0.042	0.065 ± 0.052	0.130	0.027 ± 0.039	0.035 ± 0.049	0.129
	4 h	0.000 ± 0.000	0.000 ± 0.001	0.181	0.019 ± 0.026	0.028 ± 0.037	0.197	0.010 ± 0.021	0.015 ± 0.030	0.184
	6 h	0.000 ± 0.000	0.000 ± 0.001	0.331	0.005 ± 0.006	0.010 ± 0.010	0.020	0.003 ± 0.005	0.005 ± 0.009	0.019
C_max_ (mg/dL)		0.008 ± 0.025	0.007 ± 0.010	0.829	0.121 ± 0.065	0.170 ± 0.096	0.002	0.068 ± 0.076	0.092 ± 0.108	0.007
T_max_ Median (min-max)		0.68 ± 0.38 0.50 (0.50–2.00)	0.63 ± 0.23 0.50 (0.50–1.00)	0.542	0.76 ± 0.26 1.00 (0.50–1.00)	0.71 ± 0.25 0.50 (0.50–1.00)	0.329	0.73 ± 0.32 0.50 (0.50–2.00)	0.68 ± 0.24 0.50 (0.50–1.00)	0.291
iAUC (mg·hr/dL)		0.010 ± 0.024	0.008 ± 0.014	0.774	0.254 ± 0.173	0.347 ± 0.236	0.022	0.138 ± 0.176	0.186 ± 0.241	0.029

Values are presented as mean ± SD. Abbreviation: C_max_, maximum plasma concentration; T_max_, time to reach C_max_; iAUC, incremental area under the curve. ^(1)^ Analyzed using paired *t*-test (compared between groups).

**Table 3 nutrients-13-01875-t003:** Variation in blood alcohol concentration flowing alcohol challenge test after 15 days of supplementation.

		Wild Type (*ALDH2*1/*1*)			Heterozygote (*ALDH2*2/*1*)			Total Group		
		Duolac ProAP4 Group (*n* = 19)	Placebo Group (*n* = 19)	*p*-Value ^(1)^	Duolac ProAP4 Group (*n* = 21)	Placebo Group (*n* = 21)	*p*-Value ^(1)^	Duolac ProAP4 Group (*n* = 40)	Placebo Group (*n* = 40)	*p*-Value ^(1)^
Blood alcohol level (mg/dL)	0 h	0.00 ± 0.00	0.00 ± 0.00	-	0.00 ± 0.00	0.00 ±0.00	-	0.00 ± 0.00	0.00 ± 0.00	-
	0.5 h	62.71 ± 29.85	66.27 ± 27.54	0.558	81.34 ± 31.55	77.28 ± 29.39	0.425	72.49 ± 31.79	72.05 ± 28.70	0.909
	1 h	85.54 ± 24.08	90.35 ± 23.44	0.348	90.80 ± 24.72	92.38 ± 15.98	0.750	88.30 ± 24.25	91.42 ± 19.64	0.374
	2 h	82.37 ± 10.05	81.51 ± 16.79	0.769	74.57 ± 24.19	79.40 ± 20.01	0.123	78.27 ± 19.03	80.40 ± 18.34	0.320
	4 h	49.53 ± 9.60	51.05 ± 14.72	0.511	57.16 ± 22.64	62.46 ± 20.23	0.159	53.54 ± 17.90	57.04 ± 18.52	0.116
	6 h	11.98 ± 7.79	16.69 ± 9.0	0.009	25.03 ± 13.44	31.99 ± 14.94	0.039	18.83 ± 12.81	24.73 ± 14.56	0.002
C_max_ (mg/dL)		92.39 ± 18.0	91.98 ± 21.16	0.909	94.35 ± 28.50	96.48 ± 17.89	0.673	93.42 ± 23.82	94.34 ± 19.39	0.763
T_max_ Median (min-max)		1.37 ± 0.57 1.00 (0.50–2.00)	1.18 ± 0.45 1.00 (0.50–2.00)	0.185	0.95 ± 0.31 1.00 (0.50–2.00)	1.19 ± 0.83 1.00 (0.50–4.00)	0.180	1.15 ± 0.50 1.00 (0.50–2.00)	1.19 ± 0.67 1.00 (0.50–2.00)	0.744
iAUC (mg·hr/dL)		330.11 ± 56.49	341.95 ± 81.03	0.361	359.97 ± 118.97	383.95 ± 93.41	0.127	345.79 ± 94.65	363.40 ± 89.20	0.072

Values are presented as mean ± SD. Abbreviation: C_max_, maximum plasma concentration; T_max_, time to reach C_max_; iAUC, incremental area under the curve. ^(1)^ Analyzed using paired *t*-test (compared between groups).

**Table 4 nutrients-13-01875-t004:** Variation in serum liver enzymes and blood glucose levels flowing alcohol challenge test after 15 days of supplementation.

		Wild Type (*ALDH2*1/*1*)				Heterozygote (*ALDH2*2/*1*)					Total Group					
Liver Enzymes (Standard Range)	Time	Duolac ProAP4 Group (*n* = 19)	Diff	Placebo Group (*n* = 19)	Diff	*p*-Value ^(1)^	Duolac ProAP4 Group (*n* = 21)	Diff	Placebo Group (*n* = 21)	Diff	*p*-Value ^(1)^	Duolac ProAP4 Group (*n* = 40)	Diff	Placebo Group (*n* = 40)	Diff	*p*-Value ^(1)^
AST (12~33 IU/L)	0 h	22.79 ± 4.43	-	23.47 ± 8.64	-	0.752	21.14 ± 4.52	-	20.57 ± 4.93	-	0.574	21.93 ± 4.50	-	21.95 ± 7.01	-	0.983
	1 h	22.16 ± 5.00	−0.63 ± 2.09	22.53 ± 8.12	−0.95 ± 1.99	0.672 ^(2)^	21.00 ± 3.97	−0.14 ± 2.22	21.90 ± 5.46	1.33 ± 2.03	0.032 ^(2)^	21.55 ± 4.47	−0.38 ± 2.14	22.20 ± 6.77	0.25 ± 2.30	0.217 ^(2)^
	6 h	23.53 ± 5.73	0.74 ± 2.62	23.11 ± 7.52	−0.37 ± 2.79	0.241 ^(2)^	21.48 ± 4.11	0.33 ± 1.91	21.52 ± 4.73	0.95 ± 1.99	0.374 ^(2)^	22.45 ± 4.99	0.53 ± 2.25	22.28 ± 6.18	0.33 ± 2.46	0.728 ^(2)^
ALT (5~35 IU/L)	0 h	25.42 ± 10.17	-	25.84 ± 14.66	-	0.903	24.00 ± 9.59	-	23.05 ± 10.13	-	0.598	24.68 ± 9.77	-	24.38 ± 12.4	-	0.871
	1 h	24.47 ± 10.40	−0.95 ± 2.76	25.16 ± 14.65	−0.68 ± 3.00	0.810 ^(2)^	22.00 ± 9.64	−2.00 ± 3.00	23.24 ± 9.32	0.19 ± 3.40	0.029 ^(2)^	23.18 ± 9.96	−1.50 ± 2.90	24.15 ± 12.02	−0.23 ± 3.21	0.082 ^(2)^
	6 h	24.74 ± 10.44	−0.68 ± 3.54	24.95 ± 14.19	−0.89 ± 2.81	0.864 ^(2)^	22.24 ± 9.97	−1.76 ± 2.96	22.10 ± 9.90	−0.95 ± 2.89	0.402 ^(2)^	23.43 ± 10.14	−1.25 ± 3.26	23.45 ± 12.06	−0.93 ± 2.81	0.669 ^(2)^
ALP (45~129 IU/L)	0 h	62.26 ± 11.58	-	59.53 ± 11.30	-	0.069	64.57 ± 12.27	-	59.62 ± 12.74	-	0.010	63.48 ± 14.70	-	59.58 ± 11.92	-	0.001
	1 h	62.74 ± 11.11	0.47 ± 3.13	60.89 ± 11.44	1.37 ± 2.61	0.371 ^(2)^	66.86 ± 17.24	2.29 ± 3.65	62.48 ± 13.28	2.86 ± 2.71	0.505 ^(2)^	64.90 ± 14.62	1.43 ± 3.49	61.73 ± 12.31	2.15 ± 2.73	0.259 ^(2)^
	6 h	61.84 ± 11.56	−0.42 ± 3.19	60.58 ± 11.21	1.05 ± 2.46	0.106 ^(2)^	64.05 ± 16.84	−0.52 ± 2.79	61.86 ± 13.76	2.24 ± 2.55	0.003 ^(2)^	63.00 ± 14.44	−0.48 ± 2.94	61.25 ± 12.47	1.68 ± 2.65	0.001 ^(2)^
γ-GT (12~73 IU/L)	0 h	25.05 ± 13.36	-	24.63 ± 14.01	-	0.814	17.48 ± 6.31	-	18.05 ± 5.55	-	0.616	21.08 ± 10.84	-	21.18 ± 10.84	-	0.922
	1 h	23.16 ± 12.46	−1.89 ± 2.51	23.37 ± 13.44	−1.26 ± 2.83	0.448 ^(2)^	15.33 ± 6.19	−2.14 ± 1.98	15.86 ± 4.98	−2.19 ± 3.37	0.957 ^(2)^	19.05 ± 10.35	−2.03 ± 2.22	19.43 ± 10.51	−1.75 ± 3.12	0.645 ^(2)^
	6 h	23.26 ± 11.74	−1.79 ± 3.34	23.58 ± 13.64	−1.05 ± 2.30	0.419 ^(2)^	16.67 ± 5.13	−0.81 ± 2.34	16.62 ± 5.40	−1.43 ± 3.30	0.508 ^(2)^	19.80 ± 9.39	−1.28 ± 2.86	19.93 ± 10.64	−1.25 ± 2.84	0.969 ^(2)^
Blood glucose (mg/dL)	0 h	86.00 ± 4.58	-	88.30 ± 7.41	-	0.434	83.20 ± 3.52	-	90.09 ± 7.45	-	0.015	84.53 ± 4.19	-	89.24 ± 7.30	-	0.016
	6 h	85.89 ± 4.20	−0.11 ± 5.93	84.80 ± 3.71	−3.50 ± 9.22	0.556 0.360 ^(2)^	81.0 ± 5.89	−2.20 ± 5.98	82.27 ± 4.52	−7.82 ± 9.81	0.583 0.134 ^(2)^	83.32 ± 5.61	−1.21 ± 5.88	83.48 ± 4.25	−5.76 ± 9.55	0.919 0.076 ^(2)^

Values are presented as mean ± SD. ^(1)^ Analyzed using paired *t*-test (compared between groups). ^(2)^ Analyzed using paired *t*-test (difference between change values). Abbreviation: AST, aspartate aminotransferase; ALT, alanine aminotransferase; ALP, alkaline phosphatase; γ-GT, gamma-glutamyltransferase.

## Data Availability

The datasets generated during and/or analyzed during the current study are available from the corresponding author on reasonable request.

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
