# Peer review of "Regulation of Alcohol and Acetaldehyde Metabolism by a Mixture of Lactobacillus and Bifidobacterium Species in Human"

_nutrients, 2021, doi:10.3390/nu13061875_

Round 1

Reviewer 1 Report

The manuscript by Su-Jin Jung and colleagues deals with a relevant issue on alcohol and addiction field, i.e. a novel and safe treatment to decrease acetaldehyde concentration after alcohol intake, thus decreasing alcohol-related harms. The manuscript is well written, methods are well-described and results are sound. However, I have a few indications to improve the readability and the impact of the manuscript.

  1. The abstract is not sufficiently clear/informative. The two ALDH isoforms need to be introduced. The results observed in the wild tipe group are lacking. Furthermore the statement:

    "Blood levels of alcohol and acetaldehyde in the ALDH2 heterozygote group were significantly downregulated in the probiotic-supplemented group with no changes in hangover score symptoms than the placebo group." is not clear and must be rephrased.

  2. Both introduction and discussion completely disregard the pharmacological properties of acetaldehyde in the CNS. Since acetaldehyde plays a key role in motivational properties of ethanol, I believe that the results of this study have a primary impact in the field of alcohol addiction. I suggest to review the works by M. Diana, C. Cannizzaro, M.E. Quintanilla, M. Jamal, for improving both sections.
  3. With reference to tab 5, I suggest to indicate the hangover symptoms, instead of Qs.
  4. there are a number of typos to fix in the manuscript, please read it carefully.

Author Response

 Dear reviewer 1,

Q1. The abstract is not sufficiently clear/informative. The two ALDH isoforms need to be introduced. The results observed in the wild type group are lacking. Furthermore the statement: "Blood levels of alcohol and acetaldehyde in the ALDH2 heterozygote group were significantly downregulated in the probiotic-supplemented group with no changes in hangover score symptoms than the placebo group." is not clear and must be rephrased.

Response: In response to reviewer’s recommendation, we rephrased abstract as follows:

Excessive alcohol consumption is one of the significant causes of morbidity and mortality worldwide. Alcohol is oxidized to toxic and carcinogenic acetaldehyde by alcohol dehydrogenase (ADH) and further oxidized to a non-toxic acetate by aldehyde dehydrogenase (ALDH). There are two major ALDH isoforms, cytosolic and mitochondrial, encoded by ALDH1 and ALDH2 genes, respectively. The ALDH2 polymorphism is associated with flushing response to alcohol use. Emerging evidence shows that Lactobacillus and Bifidobacterium species encode ADH and ALDH mediate alcohol and acetaldehyde metabolism, respectively. This study involves supplementation of Lactobacillus and Bifidobacterium probiotic mixture in humans and assessed their effects on alcohol and acetaldehyde metabolism. Here, twenty-seven wild types (ALDH2*1/*1) and the same number of heterozygotes (ALDH2*2/*1) were recruited for the study. The enrolled participants were randomly divided into either the probiotic (Duolac ProAP4) or the placebo group. Each group received a probiotic or placebo capsule for 15 days with subsequent crossover. Primary outcomes were measurement of alcohol and acetaldehyde in the blood after the alcohol intake. Blood levels of alcohol and acetaldehyde in the ALDH2 heterozygote group were significantly downregulated in the probiotic-supplemented group with no changes in hangover score symptoms than the placebo group. Blood levels of alcohol and acetaldehyde were significantly downregulated by probiotic supplementation in subjects with ALDH2*2/*1 genotype, but not in those with ALDH2*1/*1 genotype. However, there were no marked improvements in hangover score parameters between test and placebo groups. No clinically significant changes were observed in safety parameters. These results suggest that probiotic has a potential to downregulate the alcohol and acetaldehyde concentrations, and their effects depend on the presence or absence of polymorphism on the ALDH2 gene.

Q2. Both introduction and discussion completely disregard the pharmacological properties of acetaldehyde in the CNS. Since acetaldehyde plays a key role in motivational properties of ethanol. I believe that the results of this study have a primary impact in the field of alcohol addiction. I suggest to review the works by M. Diana, C. Cannizzaro, M.E. Quintanilla, M. Jamal, for improving both sections.

Response: We reviewed recommended articles and further described as follows:

  • In the Introduction section,

Most Caucasians have two isozymes, while approximately 30%-50% of East Asians have ALDH2 deficiency that results from the inheritance of the mutant ALDH2*2 allele [3]. Subjects with one or both alleles of ALDH2*2 experience side effects, such as facial flushing, nausea, or vomiting after the alcohol consumption As the ALDH2*2 allele has very low enzymatic activity, alcohol ingestion in subjects with one or both alleles of ALDH2*2 causes a marked elevation of blood acetaldehyde levels, which are known to generate dysphoric effects that lead to an aversion to ethanol [4].

  • In the Discussion section,

Meanwhile, our study showed that Duolac ProAP4 supplementation substantially reduced blood acetaldehyde levels but did not relieve hangover symptom scores in heterozygote and wild-type subjects. These findings are contrary to our expectation that acetaldehyde is the main contributor to the development of hangover symptoms. Under physiological conditions, venous acetaldehyde produced in the liver can’t reach the brain due to the high ALDH2 enzyme activity of the endothelial cells that line the blood brain barrier [23]. Therefore, Duolac ProAP4 may not affect acetaldehyde metabolism in the brain. However, as acetaldehyde is considered as a key player in many actions of ethanol in the brain, including behavioral changes, Duolac ProAP4 still represents a valuable therapeutic option in the management of the alcohol abuse disorders. Additionally, other factors like inflammatory cytokines, fluid imbalance, gender, ethnicity, genetic background, and nutritional status are associated with the frequency and severity of hangover symptoms along with blood acetaldehyde concentrations [24]. Thus, future studies are certainly needed to analyze the aforementioned parameters.

[23] Peana, A.T.; Sáhchez-Catalán, M.J.; Hipólito, L.; Rosas, M.; Porru, S.; Bennardini, F.; Romualdi, P.; Caputi, F.F.; Candeletti, S.; Polache, A.; Garnero, L.; Acquas, E. Mystic acetaldehyde: The never-ending story on alcoholism. Front Behav Neurosci 2017, 11, 81. 

Q3. With reference to tab 5, I suggest to indicate the hangover symptoms, instead of Qs.

Response: As suggested, we indicated hangover symptoms instead of Qs in table 5.
In contrast to the changes in alcohol and acetaldehyde concentrations, hangover symptoms were not significantly different between test and placebo groups. We therefore moved Table 5 from main text to supplementary materials (Supplementary table 1).

Q4. There are a number of typos to fix in the manuscript, please read it carefully.

Response: We have thoroughly revised our manuscript to ensure that the text is optimally phrased and free from typographical and grammatical errors.

Reviewer 2 Report

I have carefully reviewed the manuscript nutrients-1228905 with Dr. Su-Jin Jung as the first author. The title is " Regulation of alcohol and acetaldehyde metabolism by a mixture of Lactobacillus and Bifidobacterium species in human”.

The purpose of the review manuscript is to assess the capacity of Lactobacilli and Bifidobacteria to improve alcohol metabolism in a randomized, double-blind, placebo-controlled crossover human study.

This study measured blood alcohol and acetaldehyde levels in appropriate methods, and the results are considered reasonable. However, it requires a following revision prior to be acceptable for publication.

1)Because the study is cross-over design, the authors should discuss the carryover effect and the period effect in the trial.

2) The authors should add description about the standard diet in Material and Methods. Alcohol metabolism is influenced by some drinking conditions, such as with or without food ingredients.

3) Did alcohol drinking frequency during the trial period not change? It can be influenced some results due to the change. The authors should make clear it.

4) Although blood glucose level is one of the secondary outcomes, the data is not described.

5) The authors should describe how to score the subjective Alcohol Hangover Questionnaire (AHQ).

What is the range of the score? (0-maximum number)

6) I recommend that the authors reconsider the number of significant digits in the data.

7) The authors should add in the abstract that this trial was conducted as a randomized, double-blind, and placebo-controlled crossover study.

Author Response

Dear Reviewer 2, 

Q1. Because the study is cross-over design, the authors should discuss the carryover effect and the period effect in the trial.
Response: As suggested, we described the objective behind selecting crossover trial in the Method section.

2.4 Study design :

The study was designed -------------. This study was conducted as a crossover trial to reduce the differences between individuals. During the study period, between the 1st period (Day 15) and the 2nd period (Day 58), a 28-day wash-out was placed to exclude the carryover effect of the test products.

Q2. The authors should add description about the standard diet in Material and Methods. Alcohol metabolism is influenced by some drinking conditions, such as with or without food ingredients.

Response: We described the standard diet in Materials and Methods section as follows:

2.5 Alcohol challenge test

All subjects fasted for at least 12 h before the start of the trial and consumed the same standard diet (low fat diet) in the morning of the day of the clinical study. The standard diet is made up of brown rice, Chinese cabbage soup, side dishes (Korean beef stew, egg roll, and seasoned mung bean jelly salad, seasoned spinach), and sliced radish kimchi. The standard diet has about 700 Kcal. Nutritional information for the standard diet was analyzed using CAN-Pro 4.0® software (The Korean Nutrition Society, Seoul, Republic of Korea). 

Q3. Did alcohol drinking frequency during the trial period not change? It can be influenced some results due to the change. The authors should make clear it.

Response: We agree with the reviewer on influence of drinking frequency. However, to avoid the variation we specifically educated and asked subjects to refrain from drinking alcohol. Also, no significant changes on frequency were observed during the trial period. We have updated the information on this in the revised manuscript as follows:

2.4 Study design

During the study period, all subjects were educated to refrain from drinking alcohol. Subjects were educated to record the frequency of drinking alcohol consumed by themselves during the test period once a week on the alcohol consumption frequency surveys distributed in the 1st visit and 3rd visit. At the 2nd visit and 4th visit of all study subjects, an alcohol breath test was performed using a breathalyzer (ALCOFIND DA-5000, DA TECH Co., Ltd, Incheon, Korea) to confirm alcohol consumption before participating in the alcohol challenge test.

Q4. Although blood glucose level is one of the secondary outcomes, the data is not described.
Response: We updated the blood glucose levels in the revised manuscript in Table 4 and described the results as follows:

The blood glucose levels before alcohol consumption (0 h) was significantly higher in the placebo group than in the DuoLac ProAP4 group. In addition, modifications in the blood glucose levels after alcohol intake (6 h) tended to decrease more in the placebo group than in the test group. However, there was no significant difference between the groups (Table 4).

Q5. The authors should describe how to score the subjective Alcohol Hangover Questionnaire (AHQ). What is the range of the score? (0-maximum number)

Response: We described the scoring system of the subjective AHQ in the Materials and Methods section as follows:

2.6.2. Secondary outcomes

Each symptom is rated on a 5-point Likert scale ranging from 1 (no symptom) to 5 (extremely severe symptom). The total points range from 20 to 100.

Q6. I recommend that the authors reconsider the number of significant digits in the data
Response: As commented, in Table 2, the number of digits of acetaldehyde data has been changed to the third decimal place.

Q7. The authors should add in the abstract that this trial was conducted as a randomized, double-blind, and placebo-controlled crossover study.

Response: As suggested, we described study design in the Abstracts as follows:

A randomized, double-blind, placebo-controlled crossover clinical trial is designed to study the effects of Lactobacillus and Bifidobacterium probiotic mixture in humans and assessed their effects on alcohol and acetaldehyde metabolism.
